# Smooth Muscle Actin as a Criterion for Gravisensitivity of Stomach and Jejunum in Laboratory Rodents

**DOI:** 10.3390/ijms242216539

**Published:** 2023-11-20

**Authors:** Tatyana Samoilenko, Viktoriya Shishkina, Lyubov Antakova, Yelena Goryushkina, Andrey Kostin, Igor Buchwalow, Markus Tiemann, Dmitrii Atiakshin

**Affiliations:** 1Research Institute of Experimental Biology and Medicine, Burdenko Voronezh State Medical University, Moskovsky Prospekt 189a, 394036 Voronezh, Russia; antailkka@mail.ru (T.S.); 4128069@gmail.com (V.S.); tsvn@bk.ru (L.A.); goruskinaalt@mail.ru (Y.G.); 2Research and Educational Resource Center for Immunophenotyping, Digital Spatial Profiling and Ultrastructural Analysis Innovative Technologies, RUDN University, 6 Miklukho-Maklaya St, 117198 Moscow, Russia; andocrey@mail.ru (A.K.); buchwalow@pathologie-hh.de (I.B.); 3Institute for Hematopathology, 22547 Hamburg, Germany; mtiemann@hp-hamburg.de

**Keywords:** digestive system, stomach, jejunum, smooth muscle tissue, smooth muscle actin (α-SMA), space flight, antiorthostatic suspension

## Abstract

Smooth muscle tissue (SMT) is one of the main structural components of visceral organs, acting as a key factor in the development of adaptive and pathological conditions. Despite the crucial part of SMT in the gastrointestinal tract activity, the mechanisms of its gravisensitivity are still insufficiently studied. The study evaluated the content of smooth muscle actin (α-SMA) in the membranes of the gastric fundus and jejunum in C57BL/6N mice (30-day space flight), in Mongolian gerbils *Meriones unguiculatus* (12-day orbital flight) and after anti-orthostatic suspension according to E.R. Morey-Holton. A morphometric analysis of α-SMA in the muscularis externa of the stomach and jejunum of mice and Mongolian gerbils from space flight groups revealed a decreased area of the immunopositive regions, a fact indicating a weakening of the SMT functional activity. Gravisensitivity of the contractile structures of the digestive system may be due to changes in the myofilament structural components of the smooth myocytes or myofibroblast actin. A simulated antiorthostatic suspension revealed no significant changes in the content of the α-SMA expression level, a fact supporting an alteration in the functional properties of the muscularis externa of the digestive hollow organs under weightless environment. The data obtained contribute to the novel mechanisms of the SMT contractile apparatus remodeling during orbital flights and can be used to improve preventive measures in space biomedicine.

## 1. Introduction

Developing new space projects for manned flights, scientists are working hard not only to solve technical and engineering problems but also to properly maintain the health of astronauts. Weightlessness, affecting the body physiological functions and being able to cause various diseases, takes one of the leading positions in terms of the major biological effects that accompany the professional activities of the International Space Station (ISS) crews [1,2,3,4,5,6,7].

The gastrointestinal (GI) tract is critical among the vital systems of the human and animal body determining the level of adaptation to remain in extreme conditions [8]. The digestive system is highly sensitive to space flight factors, which can limit the functionality of astronauts on board the ISS [9]. As reported, significant changes occur in the digestive organs under space flight conditions [8,9,10,11], including a decreased evacuation function of the gastric contents, an increased content of gastric fluid, and an expansion of the intestine, which indicate an increased secretory activity in the organs [1]. In addition, studies have revealed structural changes in the mucosa and other layers of the stomach and intestines, resulting in a risk of progressive accumulation of changes at the tissue level up to the formation of morphological signs of atrophy and a loss of muscle mass [12,13,14,15]. All these adverse effects threaten the health and performance of the crew and the overall success of the spaceflight mission [12,16,17,18].

SMT is the major structural and functional component of all visceral organs [19,20], which, unlike skeletal muscles, is not capable of spontaneous contraction. Due to the complex mechanisms of coordination between contraction and relaxation, smooth muscles provide the movement of various entities within the visceral organs and regulate the intraorgan pressure [19,21].

SMT consists of smooth muscle cells (SMCs), which have a high level of alpha-smooth muscle actin, playing an integral role in the regulation of mechanical stress created by cells [21]. The contraction of smooth muscle tissue requires not only remodeling of the actin cytoskeleton element but also myosin activation, with further formation of the cytoskeletal network to transfer the force caused by the contractile apparatus to the extracellular matrix, as well as to adapt the SMCs to mechanical loads [21,22]. As demonstrated, not only muscle tissue cells and myocytes but also fibroblastic differon cells can express ɑ-SMA [23,24]. Due to its significant role in the functioning of the vital body systems, studies of smooth muscle tissue are of particular interest in aerospace biology and medicine. Notably, the expression of ɑ-SMA under conditions with altered gravity has not been previously investigated, which served as the rationale for this study.

## 2. Results

### 2.1. Routine Microscopy

The analysis of micropreparations (staining with Gill’s hematoxylin and eosin) involving fragments of the gastric fundus of the animals of the space flight groups helped to detect special morphological changes, which likely resulted from the impact of space flight factors on the tissue structure. The stomach walls of Mongolian gerbils (Figure 1A,B) and C57BL/6N mice normally have a typical histological structure: mucosa, submucosa, 3-component muscularis externa and serous membranes. In the samples of the first block of the study (gastric tissues of C57BL/6N mice), some structural features of the mucous membrane were found in animals of the space flight group, including changes in the integrity of the structure of the gastric glands and areas of epitheliocyte desquamation. The revealed signs can be explained by the high sensitivity of the cells of this differon to exogenous stimuli. In the muscularis externa, there was a tendency for modification in the smooth myocyte cytoarchitectonics and histotopographic rearrangements of smooth muscles at the level of functional layer formation. When analyzing samples of the stomach of animals from the BIOS-MLZH flight experiment, we recorded the preserved integrity of the glandular epithelium, and there were no alterations in the structure of the muscularis externa.

When analyzing micropreparations of the second block of the study, we detected specific morphological changes in the gastric mucosa of the Mongolian gerbils of the space flight group, probably due to the increased sensitivity of animals to stress factors, including space flight [13]. In particular, “empty” areas in the glandular epithelium with desquamated overlays were detected on the apical surface of intact cells (Figure 1C); these zones have morphological signs of necrosis. Local inflammatory infiltration was determined around these areas in the lamina propria of the mucosa. The thickness of the submucosa visually increased due to the development of perivascular edema and an increased connective tissue component in the stroma of the organ (Figure 1E). Similar signs were typical of the muscularis externa: there were layers of connective tissue strands, sometimes pronounced, between the multidirectional fibers of the muscle tissue, while normally, the endomysium has a more organized and compact structure.

When investigating the stomach of mice after antiorthostatic suspension (the third block of the study), no pronounced dystrophic changes were found in the layers of the organ wall: despite the visually detectable flattening of the glandular epithelium, there were no signs of vacuolarization and desquamation.

The jejunal wall of Mongolian gerbils (Figure 1F) and C57BL/6N mice has the following structure: mucosa, submucosa, 2-component muscularis externa and serous membranes. Normally, the mucosa due to the presence of folds, villi and crypts has a typical relief structure.

In the analysis of samples of the jejunum in Mongolian gerbils, the second block of the study (Foton-M3 experiment), the space flight group revealed morphological features typical of reactive changes: smoothing of the relief structure of the mucosa due to the appearance of a large number of cells with vacuolized cytoplasm and pyknotic nuclei. The following changes were found in the underlying structures (submucosal and muscularis externa): signs of edema were observed in the space flight group, which can be explained by the pronounced blood content of the vessels, and a proliferation of the fibrous component of the connective tissue both in the submucosa and in the intermuscular layer with an altered architectonics of muscle fiber bundles (Figure 1G).

An immunohistochemical technique was used to specify molecular rearrangements in the structure of smooth myocytes.

### 2.2. Content of α-SMA in the Stomach

The analysis of micropreparations of the first block of the study involving the material of the Bion-M1 space project detected a decrease in the immunopositive regions of ɑ-SMA expression in the orbital flight group (Table 1, Figure 2).

The morphological evaluation of α-SMA expression in the stomach wall revealed the following differences: In the groups of mice of vivarium stay and control to the readaptation group, uniform α-SMA expression was noted both in the muscularis mucosa and in the muscularis externa (Table 1, Figure 2A,D). In animals of the space flight group, a significant number of cells with weak immunopositivity to α-SMA were revealed, and their predominant number was determined in the muscularis externa (Figure 2B).

Immunopositive α-SMA cells are surrounded by endomysium, a fibrous component of the connective tissue, which accounts for approximately 10% of the total area of the muscularis externa (Table 1) and does not express α-SMA, which is normal. In the intermuscular stroma, there are nerve ganglions with neurons that do not express α-SMA. α-SMA-negative smooth muscle cells were differentiated according to specific morphological features: a spindle-shaped myocyte and an elongated nucleus of the central localization (Figure 2B,E).

In the flight group and readaptation group after the space flight, accumulations of smooth myocytes with weak immunopositivity to α-SMA were observed; their predominant number was identified in the muscularis externa of the gastric fundus of mice after exposure to weightlessness (Figure 2B,E). In most cases, these areas were found on the border of two differently directed layers of smooth muscle tissue: longitudinal—outside; circular—inside.

When analyzing α-SMA expression in the gastric mucosa, a slight trend was detected towards an increase in the immunopositive region of α-SMA—5.28% of the total area of the mucosa in the space flight group compared to 4.87% in the control group (Table 1).

There were no differences between the groups in the area of α-SMA expression in the samples obtained in a synchronous experiment in the BIOS-MLZH flight equipment model, both in the mucous membrane and in the muscularis externa.

A morphological pattern similar to the flight experiment BION-M1 was observed during the analysis of micropreparations of the second block of our study: a-SMA of Muscularis externa in Mongolian gerbils is decreased as in C57BL/N6 mice (Table 2).

The material was obtained in the flight experiment, which took place within the Foton-M3 research project. All morphological changes in microsections were also confirmed by quantitative analysis of the area of immunopositive α-SMA regions in smooth muscle tissue (Table 2). In particular, there was detected a decrease in the area of α-SMA expression in the muscularis externa of the gastric fundus of Mongolian gerbils *Meriones unguiculatus* in the space flight group compared to animals from the control group (Table 2).

Quantitative analysis of α-SMA expression in microsections of the third block of the study involving anti-orthostatic suspension revealed a different trend compared to findings detected in laboratory animals that returned from the orbital flight. No significant changes were detected in the content of α-SMA expression in the muscularis externa of the stomach of mice from the groups of control (88.51%), suspension (85%) and recovery (85.08%). An analysis of the α-SMA expression in the muscularis mucosae of mice demonstrated a slight decrease in parameters in animals from the suspension group, to 4.45% of the total area. In the vivarium control and recovery groups, the rate of immunopositivity to α-SMA was almost the same, amounting to 5.55% and 5.70%, respectively (Table 3).

### 2.3. α-SMA Expression in Structural Components of the Jejunum

The analysis of microsections of the jejunum in C57BL/N6 mice and Mongolian gerbils revealed a dynamic of the α-SMA expression similar to that detected in the stomach; it consisted of a decreased α-SMA expression after the space flight. A quantitative calculation of the area of α-SMA expression in the jejunum of C57BL/N6 mice, the first block of the study within the Bion-M1 experiment, is presented in Table 1.

An analysis of α-SMA positive cells in the jejunal mucosa of C57BL/N6 mice demonstrated increased immunopositive structures in animals from the space flight group compared to the animals from the control group (Table 1, Figure 3). This may be associated with the fact that α-SMA was expressed not only by smooth myocytes but also by fibroblastic differon cells, for example, myofibroblasts [23,25]. α-SMA positive cells are located in the connective tissue stroma of the intestinal villi; notably, the predominant number of immunopositive regions was found in animals from the space flight group (Figure 3B). This dynamic change may be due to the remodeling of the fibrous connective tissue under the impact of altered gravity factors [25].

Figure 4 demonstrates a fragment of the jejunum of the Mongolian gerbil *Meriones unguiculatus* (an experiment Foton-M 3). An Immunomorphometric analysis of muscle fibers revealed numerous clusters of immunonegative cell groups in animals of the space flight group. In animals of the control group, the α-SMA positive regions predominate (Table 2 and Figure 4).

In microsamples of the third block of the study from the experiment involving antiorthostatic suspension, no changes in the expression of α-SMA in the mucosa and muscularis externa (hereinafter in brackets, respectively) of the jejunum of C57BL/N6 mice between the control (4.61% and 86.04%), suspension (3.69% and 89.44%) and recovery groups (4.53% and 86.59%) were identified (Table 3).

Space flight factors resulted in changes in the structure of the smooth muscle tissue of the digestive system with a poorly understood molecular etiology. The results of our study evidence that weightless conditions caused a decrease in the content of α-SMA in the contractile structures of the studied hollow organs of the gastrointestinal tract. This fact was detected in Mongolian gerbils after a 12-day space flight (Table 2) and in C57BL/N6 mice after a 30-day space flight (Table 1). The data obtained can explain one of the mechanisms of the decreased functional activity of SMT in the gastric wall and jejunum after an orbital flight.

## 3. Discussion

Accumulated factual material has shown that orbital flights can lead to such changes in the gastrointestinal tract of astronauts, such as the development of hypersecretory syndrome of the stomach, the formation of venous stagnation in the vascular bed, an increase in the size of the liver, modification of the motor-evacuation function, digestive processes and activity of the pancreas and an intensification of the basal secretion of the organs of the gastroduodenal zone of astronauts or test subjects [26,27,28,29,30]. At the same time, during the flight, the astronauts noted a decrease in the feeling of thirst and appetite, a change in taste sensations, accumulation of gases in the stomach and intestines, a feeling of expansion and movement of the stomach towards the diaphragm and constipation [31]. Previously conducted morphometric studies of the muscular lining of the gastrointestinal tract in laboratory rodents returning from an orbital flight lasting 12–14 days indicate signs of involution of the smooth muscle tissue [13,14]. After the orbital flight, the content of total protein in smooth myocytes, as well as the level of RNA in the cytoplasm of smooth myocytes, significantly decreased. The data obtained were reliable not only in comparison with the indicators of animals in the vivarium control group and the synchronous experiment, but also with the antiorthostatic suspension group. The authors suggest that the observed results may be due to a decrease in the expression of the corresponding genes in the smooth myocytes of the small intestine of rats under microgravity conditions [14]. Considering that smooth myocytes actively synthesize contractile proteins, including αSMA, it can be assumed that its intracellular synthesis by smooth myocytes under microgravity conditions also decreases. The involution of smooth muscle tissue in space flight, shown in previous studies, may be the cause of the direct effect of weightlessness on smooth myocytes, which results in a decrease in the synthesis of proteins of the contractile apparatus, ryanodine receptors and other molecular components [13,14,15]. These data suggest that under conditions of weightlessness, a decrease in the motor-evacuation function of the hollow organs of the gastrointestinal tract will form both in individual smooth myocytes and at the level of the functional layers formed by them. Another important prerequisite for weakening the activity of smooth muscles in the organs of the digestive system is a significant limitation of the regulatory influence of tissue basophils on the contractile elements of the stomach and intestines due to their reduction under conditions of weightlessness [32]. This may lead to a weakening of the modulatory function of mast cells on the contraction of smooth muscle cells necessary to ensure adequate intestinal motility. The decrease in a-SMA expression revealed in our study is closely combined with previously conducted morphometric studies of the hollow organs of the digestive system and indicates a possible new reason for the modification of the peristaltic activity of the stomach and intestines of astronauts.

The balanced coordination of the contractile activity of the structural and functional elements of the gastrointestinal smooth muscles is necessary for adequate digestion; its alteration underlies a number of development disorders of gastric and intestinal motility.

Based on the results obtained in the synchronous ground experiment of the BION-M1 project in the BIOS-MLZH flight equipment model, we can conclude that the qualitative nature of the food does not affect the structural rearrangements of the smooth muscle apparatus, namely, the expression area of α-SMA.

As stated, unlike other differentiated cells, smooth myocytes are not finally determined, have plasticity and can move from contractile to proliferative, as well as pro-migrating and synthetic phenotypes [33]. Such rearrangements of SMT are characterized by a decreased expression of contractile myofilaments—actin of smooth myocytes or myofibroblasts. The switch of the SMT phenotype from contractile to synthetic is a key event for the proliferation and further migration of SMCs. However, in space biomedicine, the discovered phenotypic plasticity of SMCs should be considered in terms of changes in the ability of SMCs to remodel the extracellular matrix to form the adaptation of the digestive organs to orbital flight factors.

Our study indicates the need to further investigate the major molecular mechanisms of switching the SMT phenotype of the digestive system in both post-flight groups and in readaptation groups after space flight. In the experiment involving antiorthostatic suspension, the third block of the study, no significant changes were detected in the area of immunopositive regions of α-SMA expression; this fact may support the multidirectional impact of space flight factors and antiorthostatic suspension.

As previously demonstrated, the simulated microgravity with a random positioning apparatus resulted in a decreased content of α-SMA in fibroblast cultures, which worsened their migratory properties [34,35]. This allows for a consideration of the level of α-SMA expression as a point of application of the biological effects of weightlessness in other cells of mesodermal origin, along with smooth myocytes. Thus, simulated microgravity on cultured rat aortic smooth muscle cells using a roller culture apparatus resulted in a disorganized cytoskeleton along with suppressed proliferation and increased apoptosis [36]. Moreover, there was a decrease in the layers of smooth muscle cells along with a decreased myofilament content in the wall of the femoral arteries and anterior tibial arteries during a 4-week simulated microgravity experiment with anti-orthostatic suspension of rats [37].

Thus, as demonstrated, the gravisensitivity of smooth myocytes of the wall of the hollow gastrointestinal organs is a specific adaptive change to the altered gravity environment; that is similar to other cells of mesodermal origin with the contractile apparatus possessing α-SMA. Further research in this direction will expand the existing understanding of the mechanisms of developing organ-specific features of SMT gravisensitivity, opening up novel options for improving algorithms to prevent adverse biological effects of weightlessness.

## 4. Materials and Methods

### 4.1. Experimental Design

The study was performed in the Research Institute of Experimental Biology and Medicine, Burdenko Voronezh State Medical University. We semiquantitatively analyzed the α-SMA content in the membranes of the gastric fundus and in fragments of the jejunum in mammals: C57BL/6N mice and Mongolian gerbils *Meriones unguiculatus*; the analysis was conditionally divided into three blocks.

The first block of the study is presented by the findings obtained in a 30-day space flight on board the BION-M1 biosatellite; it involved 26 male C57BL/6N mice (19–20 weeks old, weighed 22–25 g) [38,39,40]. The group of space flight (SF) consisted of 5 animals. The group readaptation after space flight (RSF) studying the processes of recovery to the normal level of gravity included five mice; upon returning from the flight, they were kept in vivarium conditions for 7 days. These two groups of animals received paste-like food: conventional compound feed and water; casein was used as a thickener. There were two control groups (VC-SF control and VC-RSF control), including 8 animals each, for the two above groups, with standard conditions of maintenance; the animals of these groups had free access to standard feed and water during the entire experiment. A special area of research was a ground-based experiment involving 30-day simulation of selected conditions for the stay of animals on board the BION-M1 biological satellite in the BIOS-MLZH flight equipment model. The experiment included 4 groups, with 8 animals each: a baseline control group (BC), vivarium control to the group of the ground experiment (VC-BC), readaptation group examined 7 days after the ground simulation (RBC), and vivarium control to the group of 7-day readaptation after the ground simulation (VC-RBC). Detailed information on the conditions of laboratory animal maintenance in the BION-M1 experiment is presented in previous publications [40]. All procedures with animals were approved by the Commission on Biomedical Ethics, Institute of Biomedical Problems, Russian Academy of Sciences (IMBP RAS) (protocol No. 319 dated 4 April 2013).

The second block of our study involved the material obtained during the flight experiment performed within the Foton-M3 research project on the Kontur-L module [41]. Mongolian gerbils *Meriones unguiculatus* (males), aged 4–4.5 months, with an average weight of all animals of 56.2 g, were on board the spacecraft during a 12-day orbital flight [42]. The first group consisted of 12 animals that were on board the spacecraft Foton-M3 for 12 days. Upon returning from orbital flight, the animals were withdrawn from the experiment after 21 h. The second group of animals of this block of the synchronous ground experiment included 11 Mongolian gerbils. The animals were kept in the Kontur-L module, simulating space flight conditions for 12 days. Gerbils from the space flight group were placed in the Kontur-L module two days before the flight. The feed blocks consisted of natural ingredients (cereals, dried fruits, feed additives and binders with a moisture content of 20%).

Animals from three groups were subjected to euthanasia one day after the end of the relevant experiment. The average weight of the animals was about 51.6 g before the flight and about 37.1 g after the flight.

Due to a failure in the telemetry system, animals in the synchronous group were fed half the daily ration with a 4-day delay compared to the flight group (from day 9 instead of day 5). The average body weight of the animals was about 63.4 g before the experiment and about 54.2 g after the experiment [42]. The third group of animals was a vivarium control of 12 animals, which were carried out in the vivarium of the IMBP RAS. Animals were kept for two weeks in standard cages, comparable in size to cages in flight conditions; they were euthanized on the day of launch. The animals of the main group were fed the same quantity and quality of food as planned for the flight group (feed dispenser failure was not simulated in this group). Animals were not given water for 14 days under flight conditions. The average body weight of the animals before the experiment was about 53.7 g, and after the experiment, it was about 50.5 g.

All studies were carried out in compliance with the requirements for the humane treatment of animals in accordance with the decision of the Commission on Biomedical Ethics, the State Scientific Center of the Russian Federation, IMBP RAS (protocol No. 206 dated 7 October 2007).

The third block included the experimental material obtained after anti-orthostatic suspension under simulated modified gravity according to the E.R. Morey-Holton method. The simulation lasted 30 days, involved 18 C57BL/6N male mice, aged 19–20 weeks, with an average weight of 27.5 g; the animals were on a standard diet. The animals were divided into 3 experimental groups: group 1—30-day anti-orthostatic suspension (n = 6); group 2—readaptation group, the mice were exposed to 12 h support load after hanging for 30 days (n = 6); and group 3—control group, animals were kept under standard conditions (n = 6) [43]. The experimental program and all procedures with animals were approved by the Commission on Biomedical Ethics, the State Scientific Center of the Russian Federation—IMBP RAS.

### 4.2. Histoprocessing

Fragments of the gastric fundus and jejunum of the animals were fixed in 10% neutral formalin solution at room temperature, according to the standard sample preparation protocol, immediately after the animal withdrawal from the experiment. For review microscopy to study a microtome, sections were made from paraffin blocks, no more than 4-μm-thick, stained with Gill’s hematoxylin and eosin, using the picro-Mallory staining protocol.

### 4.3. Tissue Probe Staining

Immunohistochemical α-SMA detection was performed using mouse monoclonal antibodies Actin smooth muscle (1:100, Zytomed Systems, Cat. No. MSK030, Berlin, Germany) on 2 µm thick sections according to the standard protocol [44]. Homologous mouse immunoglobulins were blocked during pre-incubation of sections with unconjugated Fab fragments (goat anti-mouse IgG, Jackson ImmunoResearch, #115–007-003, 1:13 [44]. A kit with 3,3′-diaminobenzidine as a substrate was used to detect secondary antibodies conjugated with HRP (DAB Peroxidase Substrat Kit (#SK-4100), Vector Laboratories, Burlingame, CA, USA). Before the preparation embedding, the nuclei were stained with Mayer’s hematoxylin (#MHS128, Sigma-Aldrich, Darmstadt, Germany).

### 4.4. Image Acquisition

For a representative sample, a minimum of 40 visual fields were evaluated. Photodocumentation and analysis of finished micropreparations were carried out on a ZEISS Axio Imager.A2 research microscope. Quantification of α-SMA expression was determined based on the method of calculating the area of immunopositive regions using the ImageJ 1.51J8 program and expressed in absolute and relative values.

### 4.5. Statistical Analysis

The data were statistically processed using the Statistica Inc. (Tulsa, OK, USA) 12.0 package. The results were presented as mean M ± m (standard error of the mean). A Student’s *t*-test was used to assess the significance of differences between the two groups.

## 5. Conclusions

The decreased intracellular actin expression of smooth muscle cells in the muscular membrane of the stomach and jejunum after both 12-day and 30-day orbital flights of laboratory rodents indicates a high gravisensitivity of the contractile structures of the digestive hollow organs and poses new challenges for space gastroenterology. The data obtained must be considered when developing preventive measures to level the adverse biological effects of long-term space flight factors.

## Figures and Tables

**Figure 1 ijms-24-16539-f001:**
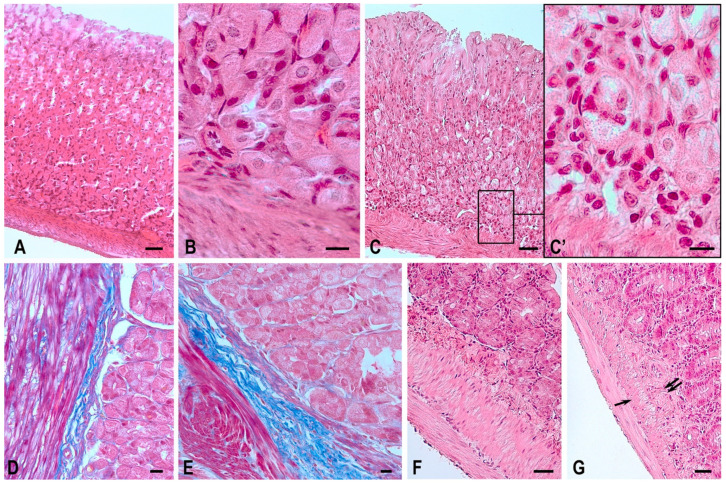
Fundus of the gastric wall and jejunum of the Mongolian gerbil Meriones unguiculatus. Technique: (**A**–**C**,**F**,**G**) Staining with hematoxylin and eosin; (**D**,**E**) Picro-Mallory staining protocol. (**A**–**E**) Morphology of the gastric wall in the control group (**A**,**B**,**D**) and in the space flight group (**C**,**C’**,**E**). (**F**,**G**) Morphology of the jejunal wall in the control group and in the space flight group (**G**). After an orbital flight occurs decreasing thickness of the (inner) circular layer muscularis externa (arrow), and proliferation of the connective tissue in the submucosa (double arrow). Scale bar: (**A**,**C**,**F**,**G**)—50 µm; (**D**,**E**)—20 µm; (**B**,**C’**)—10 µm.

**Figure 2 ijms-24-16539-f002:**
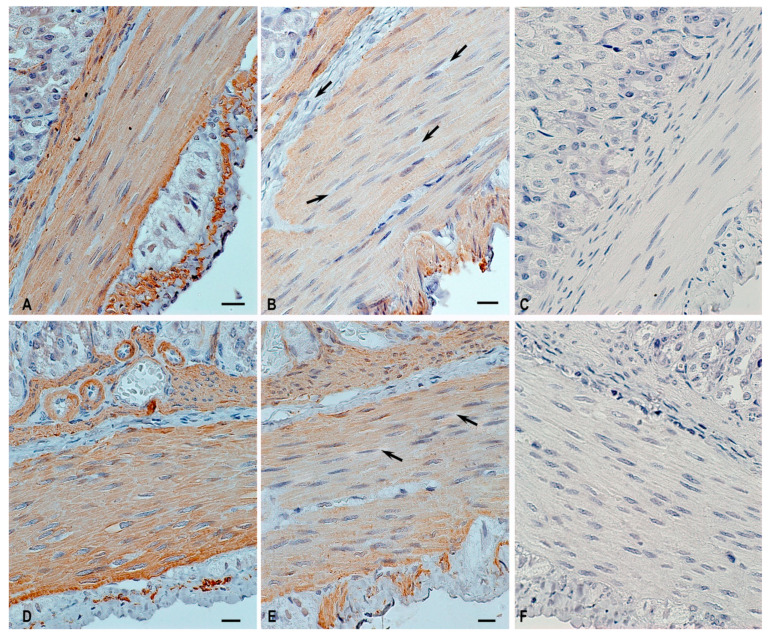
Muscularis externa of the gastric fundus in C57BL/N6 mice. Technique: (**A**,**B**,**D**,**E**) immunohistochemical α-SMA detection; (**C**,**F**) control stain (the use of primary antibodies was omitted) of vivarium control (**C**) and space flight group (**F**). (**A**)—vivarium control; (**B**)—space flight group; (**D**)—control to the readaptation group; (**E**)—readaptation after space flight. (**B**,**E**)—smooth myocytes with weak immunopositivity to α-SMA (arrow). Scale: 10 µm.

**Figure 3 ijms-24-16539-f003:**
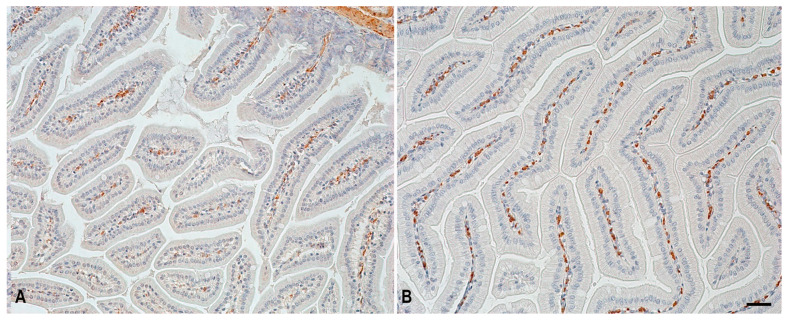
α-SMA-positive structures in the villi of the jejunal mucosa in C57BL/N6 mice. Technique: α-SMA immunohistochemical staining. (**A**)—control group; (**B**)—space flight group. Scale: 50 µm.

**Figure 4 ijms-24-16539-f004:**
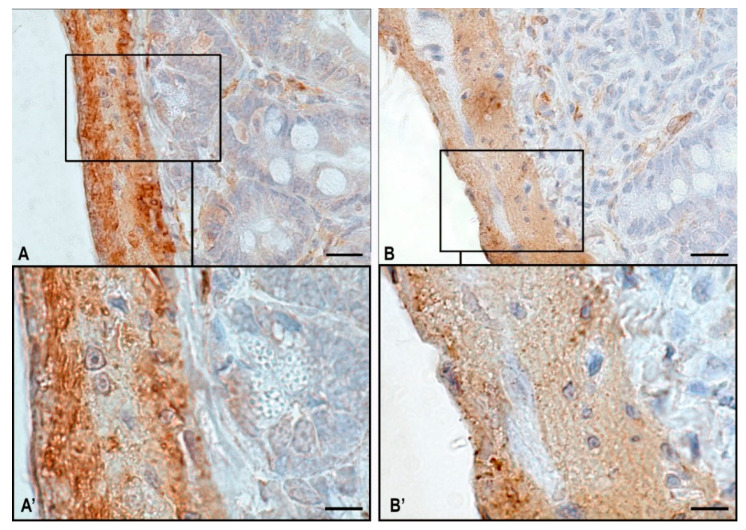
The jejunum of the Mongolian gerbil *Meriones unguiculatus* (an experiment Foton-M3). Technique: immunohistochemical α-SMA detection. (**A**,**A’**)—vivarium control; (**B**,**B’**)—space flight group. Scale: (**A**,**B**)—20 µm; (**A’**,**B’**)—10 µm.

**Table 1 ijms-24-16539-t001:** α-SMA expression in the stomach (the fundus) and jejunum in C57BL/N6 mice, BION-M1 experiment.

Groups of Animals	Mucosa			Muscularis Externa		
	Analyzed Area, mm^2^ (M ± m)	Area of α-SMA Expression, mm^2^ (M ± m)	% α-SMA	Analyzed Area, mm^2^ (M ± m)	Area of α-SMA Expression, mm^2^ (M ± m)	% α-SMA
The stomach						
VC-SF, n = 8	3.86 ± 0.04	0.19 ± 0.01	4.87	3.12 ± 0.07	2.74 ± 0.06	87.78
SF, n = 5	3.88 ± 0.09	0.20 ± 0.01	5.28 *	3.10 ± 0.06	2.32 ± 0.02 *	74.91 *
VC-RSF, n = 8	3.73 ± 0.02	0.20 ± 0.002	5.45	3.29 ± 0.02	2.81 ± 0.09	85.32
RSF, n = 5	3.72 ± 0.02	0.21 ± 0.002	5.59	2.94 ± 0.05 *	2.35 ± 0.04 *	79.79
The jejunum						
VC-SF, n = 8	3.71 ± 0.02	0.17 ± 0.002	4.50	0.57 ± 0.02	0.49 ± 0.01	85.92
SF, n = 5	3.78 ± 0.05	0.20 ± 0.01 *	5.17 *	0.58 ± 0.02	0.43 ± 0.001 *	73.50 *
VC-RSF, n = 8	3.73 ± 0.03	0.16 ± 0.01	4.42	0.60 ± 0.03	0.51 ± 0.03	84.81
RSF, n = 5	3.61 ± 0.03	0.19 ± 0.002 *	5.22 *	0.57 ± 0.03	0.46 ± 0.03	80.89

Note: *—*p* < 0.05 compared to the control.

**Table 2 ijms-24-16539-t002:** α-SMA expression in the stomach (the fundus) and jejunum of Mongolian gerbils (Foton-M3).

Groups of Animals	Mucosa			Muscularis Externa		
	Analyzed Area, mm^2^ (M ± m)	Area of α-SMA Expression, mm^2^ (M ± m)	% α-SMA	Analyzed Area, mm^2^ (M ± m)	Area of α-SMA Expression, mm^2^ (M ± m)	% α-SMA
The stomach						
Vivarium control, n = 12	3.76 ± 0.03	0.21 ± 0.002	5.67	3.59 ± 0.02	3.24 ± 0.02	90.16
Space flight, n = 12	3.55 ± 0.02	0.20 ± 0.002	5.52	3.42 ± 0.03	2.59 ± 0.04 *	75.80 *
Synchronous ground experiment, n = 11	3.54 ± 0.02	0.19 ± 0.001	5.46 *	3.57 ± 0.02	3.08 ± 0.02	86.23
The jejunum						
Vivarium control, n = 12	3.68 ± 0.02	0.26 ± 0.003	7.16	1.51 ± 0.02	1.35 ± 0.01	88.97
Space flight, n = 12	3.68 ± 0.02	0.31 ± 0.002	8.48	1.49 ± 0.01	1.13 ± 0.01	75.71 *
Synchronous ground experiment, n = 11	3.65 ± 0.01	0.26 ± 0.005	7.12 *	1.49 ± 0.02	1.24 ± 0.02	83.29

Note: *—*p* < 0.05 compared to the control.

**Table 3 ijms-24-16539-t003:** α-SMA expression in the stomach (the fundus) and jejunum in C57BL/N6 mice, anti-orthostatic suspension experiment.

Groups of Animals	Mucosa			Muscularis Externa		
	Analyzed Area, mm^2^ (M ± m)	Area of α-SMA Expression, mm^2^ (M ± m)	% α-SMA	Analyzed Area, mm^2^ (M ± m)	Area of α-SMA Expression, mm^2^ (M ± m)	% α-SMA
The stomach						
control group, n = 6	3.76 ± 0.03	0.21 ± 0.003	5.55	3.01 ± 0.04	2.66 ± 0.04	88.51
group anti-orthostatic suspension, n = 6	3.64 ± 0.06	0.16 ± 0.001 *	4.45	3.49 ± 0.06	2.97 ± 0.04 *	85 *
readaptation group, n = 6	3.76 ± 0.06	0.21 ± 0.004	5.7	3.09 ± 0.03	2.63 ± 0.04	85.08 *
The jejunum						
control group, n = 6	3.70 ± 0.04	0.17 ± 0.005	4.61	0.58 ± 0.007	0.49 ± 0.005	86.04
group anti-orthostatic suspension, n = 6	3.73 ± 0.05	0.15 ± 0.004	3.96 *	0.57 ± 0.007	0.51 ± 0.007 *	89.44 *
readaptation group, n = 6	3.73 ± 0.06	0.17 ± 0.004	4.53	0.57 ± 0.004	0.50 ± 0.005	86.59

Note: *—*p* < 0.05 compared to the control.

## Data Availability

Study data are available from the corresponding author upon reasonable request.

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
