# Peer review of "Smooth Muscle Actin as a Criterion for Gravisensitivity of Stomach and Jejunum in Laboratory Rodents"

_ijms, 2023, doi:10.3390/ijms242216539_

Round 1
Reviewer 1 Report
Comments and Suggestions for Authors
The paper of Tatyana Samoilenko et al. provides insights on role of smooth muscle tissue in gravisensitivity of GIT. It is interesting and provides potential improvement on gastrointestinal space biomedicine. However, here are some major and some minor problems to be fixed:
Point1: Considering the relationship between gravisensitivity and gastrointestinal disorder after space weightlessness, showed results of animal models should include gastrointestinal sensitivity related data such as electromyography and symptom scores.
Point 2: Figure 1 Figure 1 does not capture the entire layer of digestive tract tissue, and I can’t see the muscularis externa layer in Figure 1B. Moreover, “empty area”, “connective tissue strands”, etc. require a high magnification field of view.
Point 3: Figure 2 lacks a low magnification field of view. Moreover, the gastric muscle layers are arranged in three layers: inner oblique, middle circular, and outer longitudinal. The gastric muscle layers captured in the four images of Figure 2 are not at the same layer.
Point 4: Figure 3 requires an additional low magnification field of view.
Point 5: Group of antiorthostatic suspension lacks represent figures of gastric fundus and small intestine.
Point 6: I suggest that 2.1 and 2.2 should be described in segments based on grouping (group 1/2/3) (such as 2.1.1/2.1.2/2.1.3), and the data of the gastric fundus and small intestine should be placed separately. Currently, the groups’ results are stacked together, which is very confusing.
Point 7: Figure 3 should be modified as Figure 2 in Line 145.
Point 8: Group 3 associated table data is lacked in Line 175, 207.
Point 9: Expression of α-SMA is increased in the muscularis mucosae while decreased in muscularis externa of the gastric fundus and small intestine in Table 1 and Table 2, please explain the reason and significance in detail rather than a simple mention.
Point 10: Line 236,241, if phenotypic transformation of smooth muscle is considered, I recommend to add staining for contractile/synthetic smooth muscle cells by smMHC[PMID: 28943241], Col III[PMID: 26206426],etc. besides α-SMA staining for contractile smooth muscle cells.
Comments on the Quality of English LanguageThis paper is well-written in grammar. However, it is better to adjust the organization of results in every group in 2.1 and 2.2, which is not prominent enough.
Reviewer 2 Report
Comments and Suggestions for Authors
In this manuscript, the authors have studied the effect of gravisensitivity on smooth muscle tissue (SMT) from the stomach and jejunum of mice and gerbils.
This is potentially interesting but there are numerous issues in the paper.
Majors
1. Results, lines 73-77. In the text, it seems that in Figure 1 there are images of stomach walls of Mongolian gerbils and C57BL/6N mice but only images from Mongolian gerbils were shown. This should be better described.
2. Line 95; “The thickness of the submucosa visually increase…”, unfortunately, the image has been cut and cannot be seen well.
3. Lines 101-104. These results are not shown so in parentheses this should be indicated (e.g. data not shown). Similarly in other parts of the results (e.g. lines 157-159).
4. Table 1 and Figure 2. Why C57BL/N6 mice were studied that morphologically had no significant differences?
5. Lines 160-161; Authors stated that similar results were observed in Gerbils but Table 2 shows that a-SMA in the mucosa of gerbils is unchanged. It should be clearly indicated that the a-SMA of Muscolaris externa is decreased as in C57BL/N6 mice (Table 1).
6. It is the reviewer's opinion that the experimental group involving antiorthostatic suspension does not represent what happens in space. Orthostatic suspension eliminates the force of gravity from the suspended limbs, not the stomach.
7. Line 278-281: why did the control group receive a “standard feed” and not the food that received the space flight animals?
8. At lines 304-306 was reported the weights of gerbils but what were the weights of mice? The 30% weight reduction is notable and could indicate a general reduction in lean mass. For this reason, perhaps the reduction of a-SMA needed to be normalized with another reference protein. The results may indicate a general decline in protein.
Minors
1. The abstract is too long. About 340 words vs. 200 words indicated by IJMS instructions for the Authors. Many details are not essential in the abstract such as the vendor of the SMA antibody.
2. The lettering of Figure 4 is different compared to that of the legend.
Reviewer 3 Report
Comments and Suggestions for Authors
The manuscript remains descriptive and is limited mostly to qualitative data. The only interesting aspect is the origin of the samples from non gravidity. The title can not really be supported by results. The semiquantitative image analysis is shown in tables and should be presented as a figure e.g., by integrating them into the figures with the images. It is unclear why 2 experimental animal species were included. The experimental setting is not clear and should be depicted by a figure or graphical abstract. The discussion remains superficial and short. The weight loss of the animals at the end oft he experiments (methods section should be commented). The quality of some of the images should be improved (not sharp, e.g. Fig. 2D, Fig. 4A´B´). it would be helpful to show the subfigures in similar orientation.
The abstract contain method details which are not necessary e.g. antibody dilution and type of microscope.
Legends oft he figures: what means „fragments“ here?
Is the different thickness of the circular smooth muscle layer in the jejunum described and discussed (Fig. 1C+D)? line 120: proliferation..fibrous component. PLease indicate the changes in the figures with arrows or asterisks
Fig. 2: ganglia – does the number and size differ?
Line 75: write „to detect“
Line 80: decribe changes more specifically
Line 268: „quantitatively“ is wrong: semiquantitatively
Line 280: „standard feed“ the food oft he animal groups tob e compared differs, please discuss this limitation
Comments on the Quality of English Languageminor improvements helpful.
Reviewer 4 Report
Comments and Suggestions for Authors
The authors have studied the effect of space journey on the digestive system - smooth muscle cells of stomach and jejunum of the mice and Mongolian gerbil. The study is interesting and the experiments were well designed. The manuscript is well written. The authors concludes that orbital flights (12 days and 30 days) affect the contractile muscle of the digestive system. The study is mostly complete. However, there are several demerits in the study which needs to be addressed before its final acceptance for publication. My comments are provided below
1. The antibody used in the whole study is of mouse origin and specimen used were also from mouse origin which will increase the non-specifc signal in immunohistochemistry study and compromised the result. The authors didnot perform any negative control experiment to determine the specificity of the antibody.
2. The Figure 1A looks blury which may be due to paraffin processing of sections or low temperature of the water during section processing.
3. The authors should add a control ( section incubated with only secondary antibody) in Figure 2.
4. The authors can use the markers for the inflammatory cells in their section to determine the level of inflammation.
5. The authors also didnot use any markers for the necrosis study.
6. The proliferation of the fibrous components also needs to be shown by their respective markers for proliferation - Ki67 or PCNA.
7. The major demerits of the study is the lack any functional test. The authors may also discuss this issue.
Comments on the Quality of English LanguageThere are minor spelling mistakes in the text.
Round 2
Reviewer 2 Report
Comments and Suggestions for Authors
The paper has been revised according to my suggestions except for the point 1 and 3. As for point 1, the Authors stated “…we have clarified the text of the proposals.” Unfortunately, the text was not modified.
As for point 3, “(data not shown)” was introduced in the sentence at line 170 (new version of the manuscript) but not in the sentence at lines 102-105.
Author Response
The paper has been revised according to my suggestions except for the point 1 and 3. As for point 1, the Authors stated “…we have clarified the text of the proposals.” Unfortunately, the text was not modified.
We thank the Reviewer for his alert attention to our work. Indeed, no significant changes were made to the first sentence. However, according to the Reviewer’s recommendations, we clarified the presentation of the text of the proposal: in lines 70-72, we indicated a link to photographs of microslides of material from Mongolian gerbils only (The stomach walls of Mongolian gerbils (Figure 1A,B) and C57BL/6N mice normally have a typical histological structure: mucosa, submucosa, 3-component muscularis externa and serous membranes.).
As for point 3, “(data not shown)” was introduced in the sentence at line 170 (new version of the manuscript) but not in the sentence at lines 102-105.
Thank you for your note, we have added the phrase “(data not shown)” (line 98)
Reviewer 3 Report
Comments and Suggestions for Authors
The discussion has been improved. Some images have also been improved. However, the results are limited to
Optimize the title: „Smooth Muscle Actin as a Criterion for Gravisensitivity of a Stomach and Jejunum in Laboratory Rodents“
Remove the „a“
Based on the authors comments (referring to previous authors manuscript, thickness of muscle layers) it seems that they try to publish in little descriptive slices…
Fig. 1 B and C` should be better shown in exactly the same magnifications to allow comparison.
Line 51: „membrane“ is the wrong term (correct anatomical term)
Line 53: loss of muscle mass, the introduction would be the right place to add all the previously published data of the animal model mentioned in the authors response letter
Line 111: „inter muscular layer muscularis externa“ what does it mean? The (inner) circular layer of Tunica muscularis is shown by the arrow…The nuclear sizes suggest that F could be a slightly higher magnification than G, please check.
Line 115: serous membrane means Tunica serosa+subserosa?
Line 116: Relief structure of the mucosa – is it maintained?
Prepare the data of the tables in diagrams as proposed before
Line 290: „α- SMA“ remove surplus blank
Author Response
The discussion has been improved. Some images have also been improved. However, the results are limited to
Optimize the title: „Smooth Muscle Actin as a Criterion for Gravisensitivity of a Stomach and Jejunum in Laboratory Rodents“
Remove the „a“
The authors thank the reviewer for his comment. The title of the article is corrected.
Based on the authors comments (referring to previous authors manuscript, thickness of muscle layers) it seems that they try to publish in little descriptive slices…
Fig. 1 B and C` should be better shown in exactly the same magnifications to allow comparison.
Figure B has been replaced with a high-magnification photo, as in Figure 1C. '
Line 51: „membrane“ is the wrong term (correct anatomical term)
The authors are very grateful to the reviewer for such close attention to our work. We replaced the term "membrane" with the term "layer" on line 44.
Line 53: loss of muscle mass, the introduction would be the right place to add all the previously published data of the animal model mentioned in the authors response letter
Thanks for your note, we have added recommended sources in the "introduction" section of Line 47
Line 111: „inter muscular layer muscularis externa“ what does it mean? The (inner) circular layer of Tunica muscularis is shown by the arrow…
Thanks for the note, we have corrected the typo, instead of “inter muscular layer” we have included “The (inner) circular layer”
The nuclear sizes suggest that F could be a slightly higher magnification than G, please check.
The inaccuracy has been corrected, thanks for the note
Line 115: serous membrane means Tunica serosa+subserosa?
The reviewer is absolutely right. We have corrected the word "layers" to "structures"
Line 116: Relief structure of the mucosa – is it maintained?
Thanks to the reviewer for the important question. It escaped our attention due to the fact that the main objective of the study was devoted to the expression of α-SMA. At the same time, earlier (Atiashkin, D.A.; Bykov, E.G.; Il'in, E.A.; Pashkov, A.N. [Jejunum intersticium in Mongolian gerbils after the flight on spacecraft Foton-M3]. Aviakosm Ekolog Med 2012, 46, 8-13 .) it was shown that the total impact of orbital flight factors on the Foton-M No. 3 spacecraft led to changes in the histoarchitecture of the jejunal mucosa of Mongolian gerbils. This was evidenced by differences in the length of the villi, shortening of some of them, and the phenomenon of branching, which developed in half of the animals in the space flight group and was combined with the formation of cystic formations. It is necessary to note that the formation of cysts could occur against the background of the absence of villous branching. The most pronounced changes in this type of mucous membrane were determined in three animals. Common processes of disturbance of morphogenesis within the mucosa were absent only in two Mongolian gerbils.
In two animals, the most significant and widespread dystrophic changes in the single-layer columnar bordered epithelium were observed. In this case, in place of individual villi, only mucous tubes filled with cellular detritus could be observed, rejected into the intestinal lumen. Such changes in the villi of the mucous membrane under the influence of space flight factors were limited, since villi with an unchanged structure were simultaneously observed.
Prepare the data of the tables in diagrams as proposed before
The authors believe it might be reasonable to leave the findings for presentation. Indeed, visualizing data on a graph is more convenient for perception. However, the use of a graph allows us to reflect only the indicator “% α-SMA”, while data such as “Analyzed area, mm2” and “Area of α-SMA expression, mm2” will not be published. We appeal to the reviewer, hope for his understanding and ask him to consider our arguments for the advantage of using a tabular format to present data in this article on α-SMA expression in the stomach (the fundus) and jejunum.
Line 290: „α- SMA“ remove surplus blank
Thanks for the note, changes have been made to line 270.
Reviewer 4 Report
Comments and Suggestions for Authors
The authors have addressed all my concerns in the revised manuscript. I support the publication of the manuscript.
Comments on the Quality of English LanguageMinor spelling mistakes in the text.
Author Response
The authors thank the reviewer for his positive comment on our article and recommending it for publication.
Round 3
Reviewer 3 Report
Comments and Suggestions for Authors
my comments have been addressed.
Comments on the Quality of English Languagemy comments have been addressed.